# Metformin Treatment of Hidradenitis Suppurativa: Effect on Metabolic Parameters, Inflammation, Cardiovascular Risk Biomarkers, and Immune Mediators

**DOI:** 10.3390/ijms24086969

**Published:** 2023-04-09

**Authors:** Roisin Hambly, Niamh Kearney, Rosalind Hughes, Jean M. Fletcher, Brian Kirby

**Affiliations:** 1The Charles Centre, Department of Dermatology, St Vincent’s University Hospital, D04 T6F4 Dublin, Ireland; 2Charles Institute of Dermatology, School of Medicine, University College Dublin, D04 V1W8 Dublin, Ireland; 3School of Biochemistry and Immunology, Trinity Biomedical Sciences Institute, Trinity College Dublin, D02 R590 Dublin, Ireland; 4School of Medicine, Trinity Biomedical Sciences Institute, Trinity College Dublin, D02 R590 Dublin, Ireland

**Keywords:** hidradenitis suppurativa, HS, metformin, insulin resistance, cardiovascular risk, adipokines, CRP, metabolic syndrome

## Abstract

Hidradenitis suppurativa (HS) is a common cutaneous and systemic inflammatory disease with a significant impact on mental health and quality of life. It is associated with obesity, insulin resistance, metabolic syndrome, cardiovascular (CV) disease, and increased all-cause mortality. Metformin is used frequently in HS treatment and is effective for some patients. The mechanism of action of metformin in HS is unknown. A case-control study of 40 patients with HS (20 on metformin and 20 controls) was conducted to assess differences in metabolic markers, inflammation (C-reactive protein [CRP], serum adipokines, and CV risk biomarkers), and serum immune mediators. Body mass index (BMI), insulin resistance (77%), and metabolic syndrome (44%) were high overall, but not significantly different between the groups. This highlights the need for co-morbidity screening and management. A significant reduction in fasting insulin and a trend towards a reduction in insulin resistance were identified in the metformin group compared with pre-treatment levels. CV risk biomarkers were significantly favourable in the metformin group (lymphocytes, monocyte–lymphocyte ratio, neutrophil–lymphocyte ratio, and platelet–lymphocyte ratio). CRP was lower in the metformin group but was not statistically significant. Adipokines were dysregulated overall but were not different between the two groups. Serum IFN-γ, IL-8, TNF-α, and CXCL1 trended lower in the metformin group but did not reach significance. These results suggest that metformin improves CV risk biomarkers and insulin resistance in patients with HS. When the results of this study are considered alongside other studies in HS and related conditions, it is likely that metformin also has beneficial effects on metabolic markers and systemic inflammation in HS (CRP, serum adipokines, and immune mediators), warranting further research.

## 1. Introduction

Hidradenitis suppurativa (HS) is a cutaneous and systemic inflammatory disease characterised by recurrent painful, inflammatory lesions in the axillae and groin [1]. These can progress to scarring and sinus tract formation and can have a profound impact on quality of life [1]. Pathogenesis remains poorly understood and current evidence points to a multifactorial process including genetics, physiological/environmental factors, the microbiome, follicular occlusion, immune dysregulation, and inflammation [2]. There is a known association with obesity [3,4,5,6,7,8], insulin resistance [9,10,11], and metabolic syndrome [12,13,14,15], and these possibly play a role in pathogenesis by contributing to a systemic proinflammatory state [16,17,18]. These conditions should be screened for and addressed in the management of HS [19].

The available treatments for HS are suboptimal and include surgical options (incision and drainage, excision with primary closure, and closure by secondary intention, skin flap, or graft) [20] and a variety of medical treatments, most of which are used off-label. [21,22,23,24,25,26,27,28,29]. Adalimumab, a monoclonal antibody to TNF-α, the only licensed treatment, is approved for moderate to severe HS [30], and is effective in approximately 50% of recipients [31,32]. Metformin is used frequently in the treatment of HS. To date, there are two published studies evaluating the efficacy of metformin in HS: one prospective and one retrospective [10,26]. Efficacy has not been evaluated in a clinical trial, or by using the recommended outcome measures HS clinical response (HiSCR) [33] or the International HS Severity Score System (IHS4) [34]. The use of metformin is included in clinical guidelines under certain circumstances [20,35]. It has been suggested that metformin may be of particular benefit in mild HS, where treatment options are limited and largely consist of topical or oral antibiotics [36]. Recurrent courses of antibiotics over several years comes at a risk of antibiotic resistance [36,37,38]; therefore, it is of great clinical importance to find a viable alternative. 

Metformin is an oral biguanide licensed for the treatment of type 2 diabetes mellitus (T2DM) [39] and functions to regulate glucose metabolism [40]. It is used off-label for several diseases [41,42,43,44], including other inflammatory skin diseases [45]. It has been reported to reduce cardiovascular (CV) mortality, have a protective role against cancer and dementia, and improve longevity [40,46,47]. Metformin is an indirect agonist of adenosine monophosphate-activated protein kinase (AMPK) [48] and a mammalian target of rapamycin complex 1 (mTORC1) inhibitor [46,49]. It has been shown to alter immune cells in laboratory settings and other diseases [50,51,52,53,54,55] and may have antiandrogenic properties [56].

The precise mechanism of action of metformin in HS is uncertain. It is possible that it works through a combination of actions including reducing insulin resistance, influencing immune cells, and reducing chronic inflammation. Improved knowledge of metformin’s mechanism of action in HS may lead to an improved understanding of HS pathogenesis, and in clinical practice, it may help to identify which HS patients would benefit from metformin. 

In this paper, we report results of a case-control study comparing HS patients on metformin with those not on metformin. The primary aim was to assess differences in metabolic markers including body mass index (BMI), waist circumference, blood pressure (BP), insulin resistance, and metabolic syndrome. The secondary aims were to assess differences in inflammation as measured by C-reactive protein (CRP); CV risk biomarkers; serum cytokines and chemokines; and serum adipokines. 

We have identified some potential benefits of metformin in HS including on CV risk biomarkers and insulin resistance, and we review the potential mechanisms of action of metformin in HS based on its action in related conditions.

## 2. Results

### 2.1. Demographics

There was no statistically significant difference in gender, age, disease duration, smoking status, or Hurley stage between the HS metformin and control groups. The mean age overall was 38 years, 85% were female (*n* = 34), and the mean disease duration was 11.73 years. A total of 45% described themselves as current smokers, 22.5% as ex-smokers, and 32.5% as non-smokers. The majority (85%) had Hurley stage 2 disease. All were Caucasian (Table 1).

### 2.2. Metformin Dosing, Duration, and Response (n = 20)

A total of 3 patients were taking 500 mg daily; 4 were taking 1 g daily (500 mg twice daily); 2 were taking 1.5 g daily (500 mg three times daily or 850 mg twice daily); and 11 were taking 2 g daily (1 g twice daily). The mean duration of metformin treatment was 35.2 ± 52 months (range 6–240). Response to treatment with metformin was documented by the examining clinician as ‘partial’ in 14 (70%) and as ‘in remission’ in 6 (30%).

### 2.3. Other Medications for the Management of HS

In the metformin group (*n* = 20), 4 were taking concomitant spironolactone, 3 were taking a tetracycline and 1 was using topical treatment for the management of their HS. Five out of six who were ‘in remission’ were taking a concomitant medication (two tetracyclines, two spironolactone, and one topical treatment). Within the control group, HS medications were tetracyclines (*n* = 3), dapsone (*n* = 2), spironolactone (*n* = 1), clindamycin and rifampicin (*n* = 1), and topical treatment (*n* = 1). Four controls had previously taken metformin (more than six months prior).

### 2.4. Metabolic Parameters

There is an association between HS and obesity, metabolic syndrome, and insulin resistance [11,57], and it has been proposed that metformin may treat metabolic abnormalities in HS [10]. Comparisons of metabolic parameters were completed to identify if these were more favourable in metformin-treated patients relative to HS controls.

#### 2.4.1. BMI and Waist Circumference

The mean BMI was high overall at 34.4 kg/m^2^. A total of 25 were classified as obese (BMI ≥ 30), 10 as overweight (BMI 25–29.9), 4 as normal weight (BMI 18.5–24.9), and 1 as underweight (BMI < 18.5). A total of 10 had BMIs greater than 40: 5 in the metformin group and 5 in the control group (WHO obese class III). 

The mean waist circumference was high overall at a mean of 103 cm. A waist circumference below 94 cm is ‘low risk’, 94–102 cm is ‘high risk’, and more than 102 cm is ‘very high’ for Caucasian men [58,59]. For Caucasian women, below 80 cm is low risk, 80–88 cm is high risk, and more than 88 cm is very high risk [58,59]. Of 34 female participants, the mean waist circumference was 104.5 ± 19.3 cm, and this was significantly higher than the cut-off for ‘very high risk’ (88 cm) using a one-sample *t*-test (*p* < 0.0001). A total of 4 females had a waist circumference in the ‘low risk’ category, 5 were in the ‘high risk’ category, and 25 were in the ‘very high-risk’ category. Of 6 male participants, the mean waist circumference was lower at 94.8 ± 18.5 cm and this was not significantly different to the cut-off for ‘very high risk’ (102 cm) (*p* = 0.382). Three men were in the ‘low-risk’ category, one in the ‘high-risk’ category, and two in the ‘very high-risk’ category.

There was no statistically significant difference in BMI or waist circumference between the metformin and HS control groups (Table 1).

#### 2.4.2. Blood Pressure

Normal BP is <120/80 [60]. Systolic in the range 120–129 is classified as ‘elevated’ and BP ≥ 130/80 is classified as ‘hypertension’ [60]. Elevated BP forms one of the components for diagnosis of metabolic syndrome and is associated with increased risk of CV disease [60]. The mean BP overall in this study was elevated at 130.7/80.2, but there was no significant difference in BP between the metformin and HS control groups (Table 1).

#### 2.4.3. Insulin Resistance

Insulin resistance was calculated using HOMA-IR for 30 patients overall, with a mean of 4.46 ± 3.3. A total of 77% had a diagnosis of insulin resistance (HOMA-IR ≥ 2.6). The HOMA-IR correlated positively with both BMI (r = 0.68, *p* < 0.0001) and waist circumference (r = 0.72, *p* < 0.0001) using Spearman’s correlation. There was no statistically significant difference in mean HOMA-IR or insulin resistance between the two groups (Table 1).

Given that no difference was identified in levels of insulin resistance between HS controls and those on metformin, levels of fasting glucose, fasting insulin, and HbA1c were sought retrospectively for patients in the metformin group, prior to commencing metformin. Matched results were available on nine patients. Median fasting insulin was significantly lower on metformin than pre-metformin (22.0 mIU vs. 3.3 mIU, *p* = 0.008). The median HOMA-IR pre-metformin was 4.79 and on metformin was 3.27. This reduction was not statistically significant (*p* = 0.43). Fasting glucose and HbA1c did not differ significantly pre-metformin and on metformin (Figure 1, Table 2).

#### 2.4.4. Metabolic Syndrome

Overall, 44% of the patients were diagnosed with metabolic syndrome using NCEP/ATP III guidelines (*n* = 39). This was not significantly different between the two groups (Table 1).

### 2.5. C-Reactive Protein (CRP)

CRP is a non-specific acute-phase protein that is elevated in plasma in response to infection, inflammation, tissue damage, and malignancy [61]. It is used frequently in clinical practice as a marker of HS disease activity and severity at a given time. The median value in healthy adults is 0.8 mg/L [61]. Metformin significantly reduces CRP in women with polycystic ovarian syndrome (PCOS) [62], a disease that is associated with HS [63]. It is not yet known whether metformin has this effect on CRP in HS. 

CRP was elevated overall at 6.72 mg/L (*n* = 39) and trended lower in those treated with metformin relative to controls (8.19 vs. 5.17 mg/L). This was not statistically significant (Table 1).

### 2.6. Cardiovascular Disease Biomarkers

There is a known association between HS and CV disease [57,64], and it would be useful to be able to identify which HS patients are at increased CV risk in order to manage this appropriately. Components of the full blood count (FBC) and their ratios are reported as predictors of CV risk. Higher levels of neutrophils, platelets, and monocytes, and lower levels of lymphocytes are thought to predict higher CV risk [65,66,67,68]. Ratios calculated from the FBC are also predictors of higher CV risk: neutrophil–lymphocyte ratio (NLR), platelet–lymphocyte ratio (PLR), monocyte–lymphocyte ratio (MLR), and monocyte–HDL ratio (MHL) [69,70,71,72,73]. We analysed these CV biomarkers in 36 HS patients (17 controls and 19 on metformin) in order to identify if metformin alters CV risk. 

Absolute lymphocyte count was significantly higher in the metformin group than the HS control group (*p* = 0.037; Table 1, Figure 2). There was no statistically significant difference in neutrophil, platelet, or monocyte counts between groups (Table 1).

NLR was higher in HS controls than metformin-treated patients but did not reach statistical significance (Table 1, Figure 2). However, when compared with the general population mean [74] using a one-sample *t*-test it was significantly higher in HS controls (*p* = 0.027) but not in the metformin group (*p* = 0.094, Table 3). Similarly, PLR was higher in HS controls than in metformin-treated patients but did not reach significance (Table 1, Figure 2). It was higher in HS controls compared with the general population (*p* = 0.01) but not in metformin-treated patients compared with the general population (*p* = 0.35, Table 3) [75]. 

MLR was significantly higher in the HS control group than the metformin group (*p* = 0.032; Table 1, Figure 2). MLR was not significantly different to the mean for patients with acute coronary syndrome (ACS) in the HS control group (*p* = 0.33), but the metformin group was significantly lower than patients with ACS (*p* = 0.001, Table 3) [76]. MLR has not been studied in a general population which is why ACS was used. The monocyte–HDL ratio (MHL) was not significantly different between the two groups (Table 1, Figure 2).

None of these CV disease biomarkers correlated with CRP or HOMA-IR (Table 4).

### 2.7. Serum Cytokines and Chemokines

Metformin has been shown to inhibit expression of IL-6 and TNF-α in HaCaT keratinocytes [46,50], and to reduce IL-6 and IL-8 in rheumatoid arthritis synovial explants [51]. Metformin lowers serum TNF-α and IFN-γ in patients with T2DM [77]. Its effect on serum cytokines and chemokines in HS is not yet known. 

Serum samples were collected from 35 HS patients (18 controls and 17 metformin). Thirteen cytokines and chemokines were analysed by multiplex assay to examine whether there were differences in serum immune mediators between the two groups.

There was no statistically significant difference in levels between the two groups for IFN-γ, IL-6, IL-8, IL-17a, IL-17af, IL-18, IL-23, TNF-α, CXCL1, or CCL20, but there was a trend towards a reduction in IFN-γ, IL-8, TNF-α, and CXCL1 (Table 5, Figure 3). IL-1α, IL-1β, and IL-17c were not detected in most serum samples.

### 2.8. Adipokines

Serum adipokines are known to be dysregulated in HS [78,79]. Adiponectin is anti-inflammatory and serum levels have been shown to be decreased in HS, obesity, and psoriasis [78,79]. Resistin and leptin are both proinflammatory and their serum levels have been shown to be elevated in HS [78,79]. To examine this in our study cohort, levels of adiponectin, resistin, and leptin were analysed using ELISA in serum samples from 35 HS patients. Healthy control samples were not available in this study; however, our research group previously published a study comparing levels of these adipokines between HS and healthy controls, measured using the same technique. One-sample *t*-test was used to compare the mean level of adipokines in HS patients in our current study (*n* = 35) with the mean level of adipokines in healthy controls in our previous study (*n* = 20) [78]. Levels of adiponectin were significantly lower in HS compared with healthy controls (*p* = 0.015). Levels of leptin and resistin were significantly higher in HS compared with healthy controls (both *p* < 0.0001).

Metformin has a positive effect on adipokines in PCOS and T2DM [80,81]. The effect of metformin on adipokines in HS is not known. Levels were analysed using ELISA in serum samples from 35 HS patients (18 controls and 17 on metformin). There was no statistically significant difference between groups (Figure 4).

## 3. Discussion

Metformin is used commonly in the management of HS, either alone or in combination with other medications. It is a cheap and generally safe medication, and in clinical practice it appears to be beneficial for some patients. There is, however, very little evidence regarding its efficacy in HS and its mechanism of action in HS has not been studied to date. It has been suggested that it may work through a combination of actions including reducing insulin resistance, influencing immune cells, and reducing chronic inflammation.

This case-control study first set out to identify if there were differences in metabolic markers between HS controls and those taking metformin. Patients were representative of the known demographics of the disease and did not differ significantly between the two groups. 

BMI, waist circumference, BP, metabolic syndrome, and insulin resistance were high overall, but not significantly different between the groups. In the 40 HS patients in this study, the mean BMI was 34.4 kg/m^2^, and only 12.5% had a BMI in the normal or underweight category. Similarly, waist circumference was elevated at 103 cm, with only 4 of 34 female patients having a waist circumference in the ‘low-risk’ category. In contrast, 3 out of 6 male participants had a waist circumference in the ‘low-risk’ category. This highlights phenotypic differences between males and females with HS, as has been previously described [82,83]. Waist circumference is used as a marker of central obesity, a body fat distribution which is associated with increased cardiometabolic risk [84]. Elevated waist circumference is also one of five criteria for diagnosing metabolic syndrome as per the NCEP/APT III guidelines, of which three must be fulfilled [85]. Blood pressure was elevated overall and this represents increased CV risk [60]. HS has a known association with hypertension [63].

The prevalence of insulin resistance (HOMA-IR) was high at 77%. This is in keeping with 2 prior studies by our research group, which found a prevalence of 64% in 50 patients [9] and 75% in 36 patients [10]. These figures are higher than what has been reported in a case-control study, where insulin resistance was reported to occur in 43% of 76 HS patients compared with 16% of 61 controls [11]. PCOS is associated with insulin resistance and also with HS [86], so it is a relevant comparator in this context. The prevalence of insulin resistance in PCOS varies widely between studies, with one reporting 10.3% of lean and 31% of obese [87], and another reporting it in 70% of studied [88]. As there was no difference in insulin resistance between HS controls and those taking metformin in our current study, pre-treatment values were evaluated for those treated with metformin (available for 9 of 20). Median fasting insulin was significantly lower in HS patients taking metformin compared with their pre-treatment levels, and HOMA-IR trended lower. 

There was a 44% prevalence of metabolic syndrome, in keeping with international figures, where the prevalence is said to be 40% [89]. This is similar to the numbers reported in PCOS, with one study showing a prevalence of 43% [90].

There is a well-established significant association between HS and obesity [3,4,5,6,7,8], insulin resistance [9,10,11], metabolic syndrome [12,13,14], CV disease [57,64], T2DM [63], and all-cause mortality [64]. Insulin resistance, hormonal dysregulation, and obesity contribute to a proinflammatory state [16,17] and there is a strong correlation between the presence of insulin resistance and CV risk [91,92], as well as the risk of subsequent development of T2DM [93]. Metabolic syndrome is also an important predictor of risk of T2DM, CV disease, and all-cause mortality [94]. Taking this into account, the high prevalence of insulin resistance and metabolic syndrome in HS patients is likely to contribute to the increased all-cause mortality seen in HS. The results of our study reinforce the need to screen for and manage these important co-morbidities, and indeed this has recently been highlighted in guidelines recommending co-morbidity screening in HS [19]. Studies have shown that weight loss and bariatric surgery are associated with improvements in HS [4,95] and current guidelines incorporate weight loss advice as part of the standard management of HS [20,22,35]. These interventions should also be effective in reducing insulin resistance and metabolic syndrome [96,97], particularly as HOMA-IR correlated positively with both BMI and waist circumference. 

Metformin improves insulin resistance in people with T2DM [98]. It is recommended second line in the management for PCOS in several PCOS guidelines [99], and is sometimes used off-label to manage metabolic syndrome in obesity [100,101]. Given the association of metabolic syndrome and insulin resistance with HS, it has been considered whether metformin’s mechanism of action in HS is through improvement of insulin resistance [10]. This forms part of the rationale of treating HS patients with metformin and also raises the question of whether a patient should remain on metformin for the management of insulin resistance/metabolic syndrome/long-term CV risk, even in the absence of an improvement of their HS. In our case-control study, there was no difference in insulin resistance in 16 control HS patients compared with 14 HS patients on metformin, or in metabolic syndrome in 19 control patients compared with 20 on metformin. There was, however, a significant reduction in fasting insulin in the metformin group compared with their pre-treatment level, and a trend towards a reduction in HOMA-IR. Based on this result and what is known from other conditions, it is reasonable to conclude that metformin does have beneficial effects on insulin resistance in at least some HS patients. This should be further examined, ideally in a prospective study of HS patients commencing metformin.

Having examined the effect of metformin on metabolic parameters in HS, we next assessed the effect of metformin on inflammation, as measured by CRP. CRP was high overall indicating a high systemic inflammatory burden of the disease. CRP was higher in HS controls than those on metformin but did not reach statistical significance. This is similar to our previous finding that CRP was reduced in 41 patients taking metformin (7.9 mg/L pre-treatment to 5.2 mg/L on treatment) which trended towards significance [10]. To the best of our knowledge, the effect of metformin on CRP in HS patients has not been otherwise assessed. Metformin significantly reduces CRP in people with rheumatoid arthritis [44], PCOS [62], and T2DM [102]. Although not definitively confirmed in our current study, it is likely that metformin reduces inflammation in HS as measured by CRP and this warrants further investigation. 

HS is associated with CV disease and it would be useful to ascertain whether treatment with metformin can ameliorate this risk. In this study we compared CV disease biomarkers between HS controls and those on metformin. Higher levels of neutrophils, platelets, and monocytes, and lower levels of lymphocytes are thought to predict higher risk [65,66,67,68]. The ratios NLR, PLR, MLR, and MHR are predictors of higher CV risk [69,70,71,72,73]. We found a significant difference in some CV biomarkers in the HS metformin group compared with the HS controls, suggesting that metformin may have a cardioprotective effect in HS. Metformin has been shown to reduce NLR in patients with T2DM [103], and to reduce lymphocytes [104] and neutrophils [105] in PCOS. We previously found a reduction in these CV risk biomarkers in HS patients treated with biologic medications [106]. It would be useful to examine this effect of metformin in a larger cohort of patients. 

It is well established that cytokines, chemokines, and inflammatory proteins are abnormal in HS skin [107] and serum [108]. Systematic review identified the following immune mediators as elevated in HS serum: IL-1β, IL-6, IL-8, IL-10, IL-12p70, IL-17, TNF-α, sTNFR1, CRP, ESR, LC2, and MMP2, with conflicting results for IL-10, IL-17, and IFN-γ [109]. Metformin reduces serum inflammation and immune mediators in T2DM and may do so in other conditions associated with insulin resistance and metabolic syndrome [77,110]. The effect of metformin on serum cytokines and chemokines in HS has not previously been studied. With this in mind, 13 inflammatory cytokines and chemokines were measured from the serum of 35 HS patients using a multiplex panel for comparison between those on metformin and HS controls. There was no statistically significant difference in levels between the two groups for IFN-γ, IL-6, IL-8, IL-17a, IL-17af, IL-18, IL-23, TNF-α, CXCL1, or CCL20, but levels of IFN-γ, IL-8, TNF-α, and CXCL1 demonstrated a non-significant reduction in the metformin group. In T2DM, metformin has been shown to reduce levels of IL-6 [111], TNF-α, and IFN-γ, but not IL-17a [77], and it is plausible that the same might be seen in HS with a larger sample size. A metanalysis of the effect of metformin on IL-6 in PCOS did not find it to change significantly following treatment [62], although two studies have found reductions in serum IL-6 following metformin treatment [112].

Next, we examined adipokines in the serum of 35 HS patients and compared levels with the mean level of 20 healthy control patients in our previous study [78]. This showed that adiponectin was significantly lower in HS than healthy controls, and both leptin and resistin were significantly higher in HS than healthy controls, reinforcing that serum adipokines are abnormal in HS. Adipokines are produced by adipocytes and adipose-tissue-associated macrophages and have an important role in regulating glucose metabolism, insulin sensitivity, immunity, and inflammation [79]. Serum adipokines are known to be dysregulated in HS [78,79]. Adiponectin is anti-inflammatory and serum levels have been shown to be decreased in HS, obesity, and psoriasis [78,79]. It promotes anti-inflammatory IL-10 and suppresses TNF [113]. Resistin and leptin are both proinflammatory and their serum levels have been shown to be elevated in HS [78,79]. Leptin stimulates production of IFN-γ, IL-1β, IL-2, IL-6, and TNF as well as CCL3–5 which activates JAK2-STAT3 [113,114]. Resistin promotes expression of IL-6 and TNF-α [113]. Proinflammatory cytokines stimulate production of proinflammatory adipokines and suppress anti-inflammatory adipokines [114]. Adipokines are also abnormal in obesity, CV disease, metabolic syndrome, and T2DM [81,115], as well as chronic inflammatory conditions such as psoriasis, psoriatic arthritis, and rheumatoid arthritis [79]. 

Improving the adipokine profile in HS would likely be of benefit both in reducing cardiometabolic risk and through reducing systemic inflammation, which may in turn have a positive effect on disease burden. The effect of metformin on adipokines has not previously been investigated in HS but metformin has been shown to have a positive effect on adipokines in PCOS, metabolic syndrome, and T2DM [80,81]. Levels of adiponectin, leptin, and resistin were not different between HS controls and metformin-treated patients in our study. However, given what is known about the effect of metformin on adipokines in other conditions, it would be worth assessing adipokine levels either in a larger case-control study, or preferably prospectively when starting metformin. 

Efficacy of metformin was not evaluated in this study. However, 20 patients were taking metformin for a mean of 35 months and were noted to have either a partial or complete response to treatment, albeit some were taking concomitant medications for HS management. This indicates that for some HS patients, metformin is at least partially effective. Two published studies have examined this to date. The first demonstrated that metformin improved clinical severity of HS over 24 weeks [26]. Outcome measures were the Sartorius score which improved in 18 patients (72%) with a reduction from an average of 33.8 to 18.1 and dermatology life quality index (DLQI), which dropped significantly (more than 50%) in 64%, with a reduction from an average of 14 to 4.1. A study of 52 HS patients in our department also found metformin to be effective [10]. As this was a retrospective study, no formal severity assessment score could be used. Subjective clinical response (as assessed by the reviewing Consultant Dermatologist) was seen in 68% (36/52) with 19% (7/36) of these having quiescent disease with metformin monotherapy. A total of 75% had insulin resistance and its presence did not predict clinical response to metformin. High-quality prospective studies of metformin in HS using a recommended outcome measure (e.g., HiSCR/IHS4) would be beneficial for guiding treatment decisions. Mild HS accounts for approximately 70% of all HS cases and can have an adverse effect on quality of life, similar to more severe disease [36]. There are limited treatment options available for this cohort and recently metformin has been highlighted as potentially useful [36]. However, HiSCR and IHS4 are not practical in assessing treatment response in mild disease, so alternative outcome measures need to be considered, e.g., number of flares [36]. There is one randomised controlled trial of metformin in HS underway (ClinicalTrials.gov NCT04649502) and the results are eagerly awaited. In the era of antimicrobial resistance, finding a viable alternative to recurrent antibiotic courses for patients with mild or moderate HS is of increasing importance, and the role of metformin either as a solo or adjuvant treatment needs to be adequately assessed.

This study was limited by its case-control design, and also by missing data for some patients. A total of 20 patients were recruited to each group but some blood results were not available on all patients due to hospital laboratory processing/phlebotomy factors.

## 4. Materials and Methods

Forty HS patients were recruited from a specialty HS clinic for this case-control study: 20 who were taking metformin for at least 6 months and 20 who were not taking metformin for at least 6 months (HS controls). Patients were ≥18 years old. Patients were not taking liraglutide, other oral hypoglycaemic agents, or biologic medication (potential confounders)

The following demographic data were collected for each patient on history taking and chart review: gender; age; disease duration; smoking status; co-morbidities; metformin dose; metformin duration; concomitant medications; previous treatments for HS; and Hurley stage.

Patients were examined and the following data recorded: weight; height; waist circumference; BP; and clinical response to treatment with metformin (in remission; partial response; and no response). Body mass index was calculated.

The following blood tests were completed as per standard of care and analysed in the hospital laboratory of St Vincent’s University Hospital: full blood count (FBC); fasting lipids; fasting glucose; fasting insulin; HbA1c; and CRP. Insulin resistance was calculated using the homeostasis model of insulin resistance (HOMA-IR) with the following formula: (fasting insulin [mIU/L] × fasting glucose [mmol/L])/22.5 [116]. A value of ≥2.6 was classified as having insulin resistance, as this is the most widely used cut-off [117]. It should be noted that the cut-off for diagnosing insulin resistance may differ in different populations, e.g., by ethnicity, age, and gender [118]. Metabolic syndrome was defined according to the NCEP ATP III criteria [119,120].

A total of 13 serum cytokines and chemokines were analysed by multiplex using a custom Meso Scale Discovery (MSD, Rockville, MD, USA) U-plex™ panel (IFN-γ, IL-1α, IL-1β, IL-6, IL-8, IL-17a, IL-17af, IL-17c, IL-18, IL-23, TNF-α, CXCL1, and CCL20) according to the manufacturer’s instructions. These were chosen for their known relevance to HS pathogenesis. Levels of adiponectin, leptin, and resistin in serum were analysed using ELISA (Quantikine ELISA, R&D Systems™, Minneapolis, MN, USA) according to the manufacturer’s instructions. 

Statistical analysis was completed using SPSS (IBM Corp. Released 2020. IBM SPSS Statistics for Macintosh, Version 27.0, Armonk, NY, USA: IBM Corp) and GraphPad Prism (version 9 for Mac OS, GraphPad Software, San Diego, CA, USA). Demographic and baseline clinical data were summarised using descriptive statistics by group. Data were tested for normality and the appropriate parametric or non-parametric statistical test applied. Pearson’s Chi square was used for categorical data. For numerical data, analysis of two groups was completed using an independent *t*-test or a non-parametric equivalent (Mann–Whitney test for unpaired and Wilcoxon test for paired samples). Correlations were calculated using Pearson’s or Spearman’s correlation for parametric or non-parametric data. P values of 0.05 were considered significant.

Ethics approval was granted by the St Vincent’s Healthcare Group Research Ethics Committee (reference number RS18-063) and informed written consent was obtained from all participants.

## 5. Conclusions

Our findings indicate a high prevalence of metabolic co-morbidities in HS patients as well as elevated CRP and serum adipokines, highlighting the systemic burden of the disease and the need to diagnose and treat these co-morbidities. Our results suggest that metformin improves CV risk biomarkers and fasting insulin in HS. The effect on insulin resistance, metabolic syndrome, serum immune mediators, and serum adipokines are less clear based on this study alone; however, when our results are considered alongside results from other studies in HS and related conditions, it is likely that metformin has beneficial effects on metabolic markers and systemic inflammation in HS patients. Given that patients with HS have increased all-cause mortality, a medication that can improve cardiometabolic risk and systemic inflammation in this condition would be very welcome. This warrants further clinical and translational research; ideally, a prospective study of patients commencing metformin should be conducted with analysis of metabolic markers, inflammation (CRP), CV risk biomarkers, serum immune mediators, and adipokines, as well as skin immune mediators at baseline and after 6 months of treatment. A prospective study would also allow an assessment of efficacy using a recommended outcome measure such as HiSCR and/or IHS4. This would help to guide treatment decisions in the management of HS and allow a more evidence-based approach to treatment. Mechanistic studies examining the effect of metformin on HS skin either in vitro or ex vivo would be of complementary benefit and represent an important next step to further understanding the role of metformin in treating this debilitating condition.

## Figures and Tables

**Figure 1 ijms-24-06969-f001:**
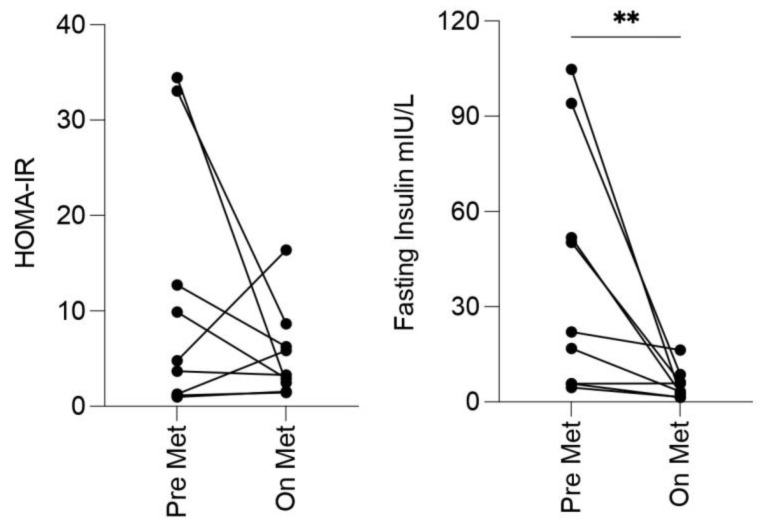
Fasting insulin levels were lower in patients taking metformin compared with their pre-treatment level. Blood samples from HS patients taking metformin pre-treatment and on treatment (*n* = 9) were analysed for HOMA-IR and fasting insulin. Graphs represent matched samples. Statistical significance was calculated using the Wilcoxon test. ** *p* ≤ 0.01. HOMA-IR = homeostatic model assessment for insulin resistance; Met = metformin.

**Figure 2 ijms-24-06969-f002:**
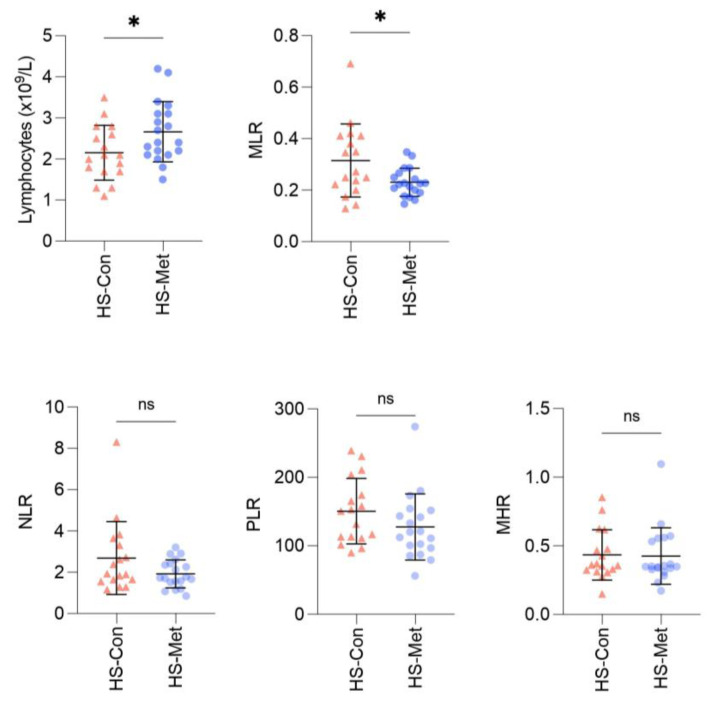
The absolute number of lymphocytes was higher in HS patients taking metformin than HS controls and the monocyte–lymphocyte ratio (MLR) was lower in those taking metformin than controls. Blood samples from control HS patients (*n* = 17) and those taking metformin (*n* = 19) were analysed for the absolute number of lymphocytes, the monocyte to lymphocyte ratio (MLR), neutrophil–lymphocyte ratio (NLR), platelet–lymphocyte ratio (PLR), and monocyte–HDL ratio (MHL). Graphs represent individual results, mean and standard deviation. Statistical significance was calculated using independent samples *t*-test. * *p* < 0.05. ns = not significant. MHR = monocyte–HDL ratio (HDL = high-density lipoprotein); MLR = monocyte–lymphocyte ratio; NLR = neutrophil–lymphocyte ratio; and PLR = platelet–lymphocyte ratio.

**Figure 3 ijms-24-06969-f003:**
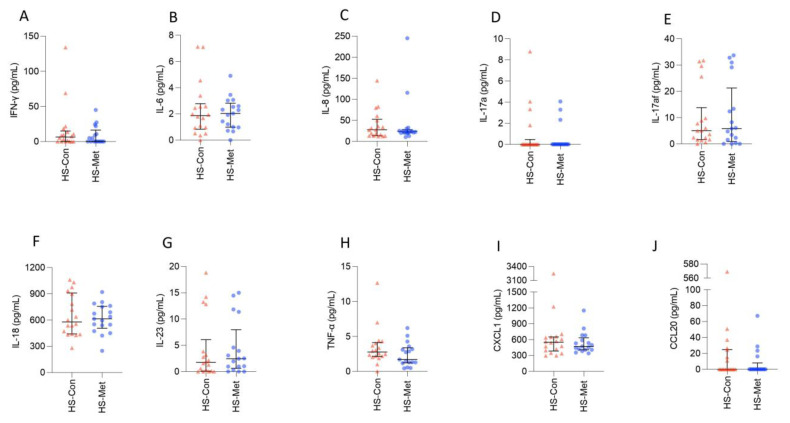
Trend towards a reduction in serum concentration of IFN-γ, IL-8, TNF-α, and CXCL1 in HS patients taking metformin compared with HS controls. Serum levels of cytokines and chemokines from control HS patients (*n* = 18) and those taking metformin (*n* = 17) were analysed using a multiplex assay: IFN-γ (**A**), IL-6 (**B**), IL-8 (**C**), IL-17a (**D**), IL-17af (**E**), IL-18 (**F**), IL-23 (**G**), TNF-α (**H**), CXCL1 (**I**), and CCL20 (**J**). Graphs represent individual samples with median and IQR. Statistical significance was calculated using the Mann–Whitney test. IQR = interquartile range.

**Figure 4 ijms-24-06969-f004:**
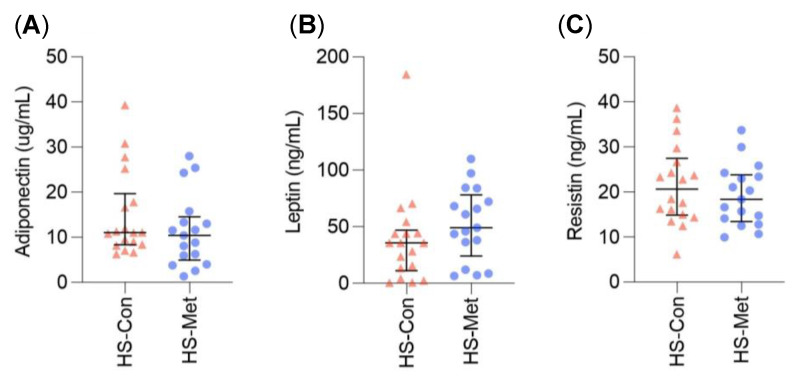
No statistically significant difference in serum adipokines between HS controls (HS-Con) and those on metformin (HS-Met). Serum levels of adipokines adiponectin (**A**), leptin (**B**), and resistin (**C**) were analysed using ELISA in 18 HS controls and 17 HS patients taking metformin. Graphs represent individual samples with median and IQR. Statistical significance was calculated using the Mann–Whitney test. IQR = interquartile range.

**Table 1 ijms-24-06969-t001:** Demographics, metabolic parameters, CRP, and CV risk biomarkers for HS metformin and HS controls groups.

	HS Metformin (*n* = 20)	HS Control (*n* = 20)	*p*-Value
Female (*n*, %)	19 (95%)	15 (75%)	0.077
Age (mean ± SD)	38.1 ± 11.2	37.95 ± 12.3	0.968
Disease duration (mean ± SD)	12.65 years ± 8.7	10.8 years ± 8.6	0.504
Smoking (*n*, %)			0.374
Current	7 (35%)	11 (55%)	
Ex	6 (30%)	3 (15%)	
Non	7 (35%)	6 (30%)	
Hurley stage (*n*, %)			0.347
Hurley 1	2 (10%)	2 (10%)	
Hurley 2	16 (80%)	18 (90%)	
Hurley 3	2 (10%)	0	
BMI kg/m^2^ (mean ± SD)	35 ± 9.8	33.6 ± 9	0.624
WC cm (mean ± SD)	103.7 ± 19	102.35 ± 19.8	0.823
SBP (mean ± SD)	131 ± 14.9	130.5 ± 18.8	0.528
DBP (mean ± SD)	81.3 ± 9	79.2 ±12.3	0.116
HbA1c mmol/mol (mean ± SD)	36.2 ± 2.7	35.8 ± 4.3	0.74
HOMA-IR (mean ± SD)	4.7 ± 4 (*n* = 14)	4.2 ± 2.8 (*n* = 16)	0.673
Insulin resistance (HOMA-IR ≥ 2.6) (*n*, %)	10 (71.4%) (*n* = 14)	13 (81.3%) (*n* = 16)	0.526
Metabolic syndrome (*n*, %)	9 (45%) (*n* = 20)	8 (42%) (*n* = 19)	0.855
CRP (mg/L) (mean ± SD)	5.17 ± 3.8 (*n* = 19)	8.19 ± 11.9 (*n* = 20)	0.3
	*n* = 19	*n* = 17	
Neutrophils (×10^9^/L) (mean ± SD)	4.88 ± 1.5	5.15 ± 2	0.658
Lymphocytes (×10^9^/L) (mean ± SD)	2.66 ± 0.73	2.15 ± 0.67	0.037
Platelets (×10^9^/L) (mean ± SD)	314.16 ± 56	300.65 ± 57.4	0.48
Monocytes (×10^9^/L) (mean ± SD)	0.59 ± 0.12	0.62 ± 0.18	0.596
NLR (mean ± SD)	1.92 ± 0.68	2.69 ± 1.76	0.108
PLR (mean ± SD)	127.65 ± 48.2	150.6 ± 47.74	0.161
MLR (mean ± SD)	0.23 ± 0.05	0.32 ± 0.14	0.032
MHR (mean ± SD)	0.43 ± 0.21	0.43 ± 0.18	0.898

Statistical significance was calculated using Pearson’s Chi-square for categorical data and independent samples *t*-test for numerical data. BMI = body mass index; CRP = C-reactive protein; CV = cardiovascular; DBP = diastolic blood pressure; HOMA-IR = homeostatic model assessment for insulin resistance; HS = hidradenitis suppurativa; MHR = monocyte–HDL ratio (HDL = high-density lipoprotein); MLR = monocyte–lymphocyte ratio; *n* = number; NLR = neutrophil–lymphocyte ratio; PLR = platelet–lymphocyte ratio; SBP = systolic blood pressure; SD = standard deviation; and WC = waist circumference.

**Table 2 ijms-24-06969-t002:** Fasting insulin levels were lower in patients taking metformin compared with their pre-treatment level.

	Pre-Metformin (*n* = 9)	On Metformin (*n* = 9)	*p*-Value
Fasting glucose mmol/L (median, IQR)	4.9 (4.6–5.3)	4.9 (4.8–5.3)	0.69
Fasting insulin mIU/L (median, IQR)	22.0 (5.7–72.9)	3.3 (2.0–7.5)	0.008
HOMA-IR (median, IQR)	4.8 (1.2–22.9)	3.3 (2.0–7.5)	0.43
HbA1c mmol/mol (median, IQR)	35.0 (32.5–37.8)	36.0 (35.0–38.0)	0.11

Statistical significance was calculated using the Wilcoxon test for non-parametric data. IQR = interquartile range.

**Table 3 ijms-24-06969-t003:** NLR and PLR were higher in the HS control group than the general population mean but not significantly different in the HS metformin group compared with the general population mean. MLR was lower in the metformin group than the ACS mean.

CV Risk Biomarker	Group	Mean	*p*-Value
NLR	HS Control (*n* = 17)	2.69	0.027
	General population mean ^a^	1.65	
	HS Metformin (*n* = 19)	1.92	0.094
	General population mean	1.65	
PLR	HS Control (*n* = 17)	150.6	0.01
	General population mean ^a^	117.05	
	HS Metformin (*n* = 19)	127.65	0.35
	General population mean ^a^	117.05	
MLR	HS Control (*n* = 17)	0.32	0.325
	ACS mean ^a^	0.28	
	HS Metformin (*n* = 19)	0.23	0.001
	ACS mean ^a^	0.28	

^a^ The general population mean and ACS mean were identified from the literature (references in text). Statistical significance was calculated using one-sample *t*-test. ACS = acute coronary syndrome; CV = cardiovascular; NLR = neutrophil–lymphocyte ratio; PLR = platelet–lymphocyte ratio; and MLR = monocyte–lymphocyte ratio.

**Table 4 ijms-24-06969-t004:** CV biomarkers did not correlate with CRP or HOMA-IR.

Correlation	r	*p*-Value
CRP and lymphocyte count	0.106	0.546
CRP and neutrophil count	0.179	0.304
CRP and NLR	0.023	0.896
CRP and PLR	0.052	0.766
CRP and MLR	−0.084	0.631
CRP and MHR	0.218	0.208
HOMA-IR and lymphocyte count	−0.4	0.841
HOMA-IR and neutrophil count	0.209	0.296
HOMA-IR and NLR	0.091	0.653
HOMA-IR and PLR	0.068	0.763
HOMA-IR and MLR	−0.035	0.861
HOMA-IR and MHR	0.157	0.435

Calculated using Pearson’s correlation. n = 35 for CRP calculations; n = 27 for HOMA-IR calculations. CRP = C-reactive protein; CV = cardiovascular; HOMA-IR = homeostatic model assessment for insulin resistance; MHR = monocyte–HDL ratio; MLR = monocyte–lymphocyte ratio; NLR = neutrophil–lymphocyte ratio; and PLR = platelet–lymphocyte ratio.

**Table 5 ijms-24-06969-t005:** Trend towards a reduction in serum concentration of IFN-γ, IL-8, TNF-α, and CXCL1 in HS patients taking metformin compared with HS controls.

Analyte	HS Controls *n* = 18 Median (IQR)	HS Metformin *n* = 17 Median (IQR)	*p*-Value
IFN-γ (pg/mL)	6.43 (0–15)	0.19 (0–16.34)	0.52
IL-6 (ng/mL)	1.9 (0.84–2.78)	2 (0.99–2.81)	0.8
IL-8 (ng/mL)	27.39 (13.34–52.44)	24.03 (20.05–27.75)	0.81
IL-17a (pg/mL)	0 (0–0.45)	0 (0–0)	0.74
IL-17af (pg/mL)	5.06 (1.65–13.77)	5.83 (0.93–21.24)	0.98
IL-18 (pg/mL)	578 (441–908)	613 (506–757)	0.96
IL-23 (pg/mL)	1.75 (0.09–6.08)	2.46 (0.63–7.97)	0.68
TNF-α (pg/mL)	2.78 (2.15–4.1)	1.7 (1.22–3.38)	0.19
CXCL1 (ng/mL)	553 (384–652)	471 (408–635)	0.73
CCL20 (pg/mL)	0 (0–24.89)	0 (0–8.11)	0.37

Statistical significance was calculated using the Mann–Whitney test. IQR = interquartile range.

## Data Availability

The data that support the findings of this study are available from https://figshare.com/articles/dataset/Data_supporting_findings_of_study/22575025.

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
