# Peer review of "Metformin Treatment of Hidradenitis Suppurativa: Effect on Metabolic Parameters, Inflammation, Cardiovascular Risk Biomarkers, and Immune Mediators"

_ijms, 2023, doi:10.3390/ijms24086969_

Round 1

Reviewer 1 Report

Thank you very much for the opportunity to read this very interesting manuscript. Hidradenitis suppurativa and insulin resistance is a very common problem and the evaluation of metformin treatment is very important for the daily practice of doctors treating HS. This is a unique study and an interesting one. And the results are interesting. The manuscript is written in a distinctive form, the language is very good, sounding scientific and the structure of such paper is typical for this type of manuscripts.

I have some minor suggestions : 

1) Please elaborate on any abbreviations under figures and tables (legends) e.g. table 1 table 2 etc. under some figures abbreviations are there but this is not the case everywhere

2) Conclussions - in my opinion you should separate such a subsection and summarise the results. Not as an extension of the discussion but a summary in Conclussions.

3) Introduction Section - in my opinion some information about the characteristics of the disease is missing i.e. there is no sentence about how the disease affects the quality of life - suggests PMID : 36686013

Yet still in the world the 1st line of treatment is antibiotic therapy , this results in increased multidrug resistance of bacteria that colonise HS lesions.  PMID : 36686013

I think it is worth highlighting this given looking at the global problem of superbugs, this will also significantly increase the conclusions about metformin treatment. Perhaps we need to treat adjuvantly with metformin as standard ? This will perhaps avoid in the future the overproduction of drug-resistant strains that are the result of antibiotics overtreatment in HS. 

4) Introduction Section - Biological treatment is described here, but there is at least one sentence missing about the surgical methods available for the treatment of HS - this is very important as in severe cases surgery gives very good results. I suggest that the authors include information on surgical treatment of HS in the introduction section. 

PMID : 36686007

PMID : 36004913

Congrats on an interesting paper which is very relevant. I think the authors will address any suggestions well.

Best regards,

Author Response

Thank you very much for the opportunity to read this very interesting manuscript. Hidradenitis suppurativa and insulin resistance is a very common problem and the evaluation of metformin treatment is very important for the daily practice of doctors treating HS. This is a unique study and an interesting one. And the results are interesting. The manuscript is written in a distinctive form, the language is very good, sounding scientific and the structure of such paper is typical for this type of manuscripts.

Thank you for taking the time to review our paper and for your suggestions, which we have addressed below.

I have some minor suggestions : 

1) Please elaborate on any abbreviations under figures and tables (legends) e.g. table 1 table 2 etc. under some figures abbreviations are there but this is not the case everywhere.

These have been added where missing.

2) Conclusions - in my opinion you should separate such a subsection and summarise the results. Not as an extension of the discussion but a summary in Conclusions.

The final paragraph of the discussion section has been moved to the conclusion section.

3) Introduction Section - in my opinion some information about the characteristics of the disease is missing i.e. there is no sentence about how the disease affects the quality of life - suggests PMID : 36686013

 The first sentence of the introduction has been expanded as follows: Hidradenitis suppurativa (HS) is a cutaneous and systemic inflammatory disease characterised by recurrent painful, inflammatory lesions in the axillae and groin.1These can progress to scarring and sinus tract formation, and can have a profound impact on quality of life.1

Yet still in the world the 1st line of treatment is antibiotic therapy , this results in increased multidrug resistance of bacteria that colonise HS lesions.  PMID : 36686013

I think it is worth highlighting this given looking at the global problem of superbugs, this will also significantly increase the conclusions about metformin treatment. Perhaps we need to treat adjuvantly with metformin as standard ? This will perhaps avoid in the future the overproduction of drug-resistant strains that are the result of antibiotics overtreatment in HS. 

 This is an important point. The following sentence has been added to the introduction section “ It has been suggested that metformin may be of particular benefit in mild HS, where treatment options are limited and largely consist of topical or oral antibiotics.33 Recurrent courses of antibiotics over several years comes at a risk of antibiotic resistance33-35  and it is of great clinical importance to find a viable alternative.”

The following sentence has been added to the discussion: In the era of antimicrobial resistance, finding a viable alternative to recurrent antibiotic courses for patients with mild or moderate HS is of increasing importance, and the role of metformin either as a solo or adjuvant treatment needs to be adequately assessed.

4) Introduction Section - Biological treatment is described here, but there is at least one sentence missing about the surgical methods available for the treatment of HS - this is very important as in severe cases surgery gives very good results. I suggest that the authors include information on surgical treatment of HS in the introduction section. 

PMID : 36686007

PMID : 36004913

 The following sentence has been included in the introduction.  “Surgical treatments including incision and drainage, excision with primary closure, closure by secondary intention, skin flap or graft, are utilized frequently in HS management.32

Thank you for the suggested references, where possible we have utilised references from guidelines, which were already included in our paper. We have included one of your suggested references (antibiotic study).

Congrats on an interesting paper which is very relevant. I think the authors will address any suggestions well.

Best regards,

Reviewer 2 Report

Roisin Hambly et al. herein present a paper well-written and fluently readable. 

The topic is of interest, and the role of Metformin on HA, although already investigated ad debated, deserves to be further clarified.   

Minor suggestions:

Fig 1 could be improved by adding regression analysis

Discussion should be shortened or perhaps divided into discussions and conclusions. 

Author Response

Roisin Hambly et al. herein present a paper well-written and fluently readable. 

The topic is of interest, and the role of Metformin on HA, although already investigated ad debated, deserves to be further clarified.   

Thank you for taking the time to review our paper and for your suggestions, which we have addressed below.

Minor suggestions:

Fig 1 could be improved by adding regression analysis

Thank you for this suggestion. We have sought advice regarding this and given the n of 9, it was concluded that regression analysis would be of limited additional value to the analysis already presented.

Discussion should be shortened or perhaps divided into discussions and conclusions. 

This section has been divided into discussion and conclusion for clarity

Reviewer 3 Report

The issue addressed by the study is a significant one and the data useful additions to the literature.  The authors note the limitations of the study design, number of patients involved, etc. and the constraints on drawing clear conclusions from the data.

Author Response

The issue addressed by the study is a significant one and the data useful additions to the literature.  The authors note the limitations of the study design, number of patients involved, etc. and the constraints on drawing clear conclusions from the data.

Thank you.

Round 2

Reviewer 1 Report

Accept in present form 

Author Response

There were no comments to address.